

# Complete plastid genome sequences of two species of the Neotropical genus *Brunellia* (Brunelliaceae)

Janice Valencia-D[1], José Murillo-A[2], Clara Inés Orozco[2], Carlos Parra-O[2] and Kurt M. Neubig[1]

[1] School of Biological Sciences, Southern Illinois University at Carbondale, Carbondale, IL, United States of America
[2] Instituto de Ciencias Naturales, Universidad Nacional de Colombia, Bogotá D.C., Colombia

## ABSTRACT

Here we present the first two complete plastid genomes for Brunelliaceae, a Neotropical family with a single genus, *Brunellia*. We surveyed the entire plastid genome in order to find variable cpDNA regions for further phylogenetic analyses across the family. We sampled morphologically different species, *B. antioquensis* and *B. trianae*, and found that the plastid genomes are 157,685 and 157,775 bp in length and display the typical quadripartite structure found in angiosperms. Despite the clear morphological distinction between both species, the molecular data show a very low level of divergence. The amount of nucleotide substitutions per site is one of the lowest reported to date among published congeneric studies ($\pi = 0.00025$). The plastid genomes have gene order and content coincident with other COM (Celastrales, Oxalidales, Malpighiales) relatives. Phylogenetic analyses of selected superrosid representatives show high bootstrap support for the ((C,M)O) topology. The N-fixing clade appears as the sister group of the COM clade and Zygophyllales as the sister to the rest of the fabids group.

## INTRODUCTION

*Brunellia* Ruiz & Pav., with about 60 species, is the only genus within the Neotropical family Brunelliaceae Engl. Its range of distribution is from southern Mexico to Bolivia, and a single species reaches the Greater Antilles (*Orozco et al., 2017*). They are evergreen trees, some of which can reach 40 m in height and 1 m DBH in high mountain forests, or grow less than 10 m high at over 3,600 m of elevation (*Cuatrecasas, 1970*). They are also important elements of the Andean ecosystems where they have diversified, and comprise part of the high endemism of these zones (*Orozco, 2001*). The narrow range of distribution of the vast majority of the species make *Brunellia* an interesting group to study from phylogenetic and biogeographical perspectives.

Brunelliaceae belong to the order Oxalidales in a clade formed with Cephalotaceae, Cunoniaceae and Elaeocarpaceae (*Zhang & Simmons, 2006*; *Soltis et al., 2011*; *Heibl & Renner, 2012*; *Sun et al., 2016a*; *Li et al., 2019*). Oxalidales belong to the commonly called

Corresponding author
Janice Valencia-D,
janice.valenciaduarte@siu.edu

'COM clade' together with Celastrales and Malpighiales (*Matthews & Endress, 2006*), and it has been broadly recognized as monophyletic based on plastid data (but see *Zhao et al., 2016*; *Zeng et al., 2017*; *Leebens-Mack et al., 2019*). However, despite support for monophyly of each of these three orders, there is still some controversy regarding the relationships among them.

For Brunelliaceae, the first phylogenetic analysis was made by *Orozco (2001)* based on morphological characters. In that study, *Brunellia* was recognized as monophyletic with five morphological characters as synapomorphies. *Orozco (2001)* proposed five sections for the genus, but the phylogenetic tree had poor resolution and low support. A phylogenetic analysis of *Brunellia* using molecular characters, both from nuclear and plastid DNA is currently ongoing (José Murillo-A, Clara Inés Orozco, Carlos Parra-O, Alvaro J. Perez and Katya Romoleroux, 2019, unpublished data). In a preliminary survey, plastid DNA regions (*atpB-matK, ndhF, psaB-rps14, psaI-accD, psbJ-petA, psbA-trnH, rbcL, trnS-trnG*) were used, but they showed low sequence variation, making them unsuitable for resolving relationships within this genus.

With the purpose of identifying variable cpDNA regions, we surveyed the entire plastid genome of two morphologically different species, *Brunellia antioquensis* (Cuatrec.) Cuatrec. and *B. trianae* Cuatrec., which belong to the biggest sections in the genus, Sect. *Brunellia* and Sect. *Simplicifolia*, respectively. *Brunellia antioquensis* has an ochraceous to fulvous pubescence, stipules 3–5 mm long, compound leaves and fruits with U-shaped endocarp, whereas *B. trianae* has an appressed, lanate to arachnoid, ochraceous indument, stipules 0.5–2 mm long, simple leaves and fruits with boat-shaped endocarp (Fig. 1).

Here we explore the variability and utility of the plastid DNA in Brunelliaceae and document for first-time complete cp-genome sequences for two species. We characterize the plastid genome of each species and compare them in terms of divergence hotspots in coding and non-coding regions. We also analyze the plastid gene organization of *Brunellia* and ten other COM representatives. Finally, using 75 protein-coding regions available for 41 selected superrosid species; we establish the phylogenetic position of Brunelliaceae and produce a hypothesis of relationships among the COM clade and related orders.

## MATERIALS & METHODS

### DNA sampling, extraction and sequencing

The samples used in this study were collected under the institutional Universidad Nacional de Colombia collection permit (Number: 0255 March 2014). Collections were made in Colombia in Cerro del Padre Amaya, Antioquia department, rural areas of Medellín, in October 2012 (*B. antioquensis C.I. Orozco 4001*, *B. trianae C.I. Orozco 4015*). Sections of leaves were dried on silica gel. Vouchers specimens of both collections were deposited at the Herbario Nacional Colombiano (COL). These species belong to two different clades that were identified using ITS and ETS regions (José Murillo-A, Clara Inés Orozco, Carlos Parra-O, Alvaro J. Perez and Katya Romoleroux, pers. obs., 2019).

DNA was extracted using a CTAB method (*Doyle & Doyle, 1987*), followed by a silica purification column step and elution in Tris-EDTA (*Neubig et al., 2014*). DNA samples

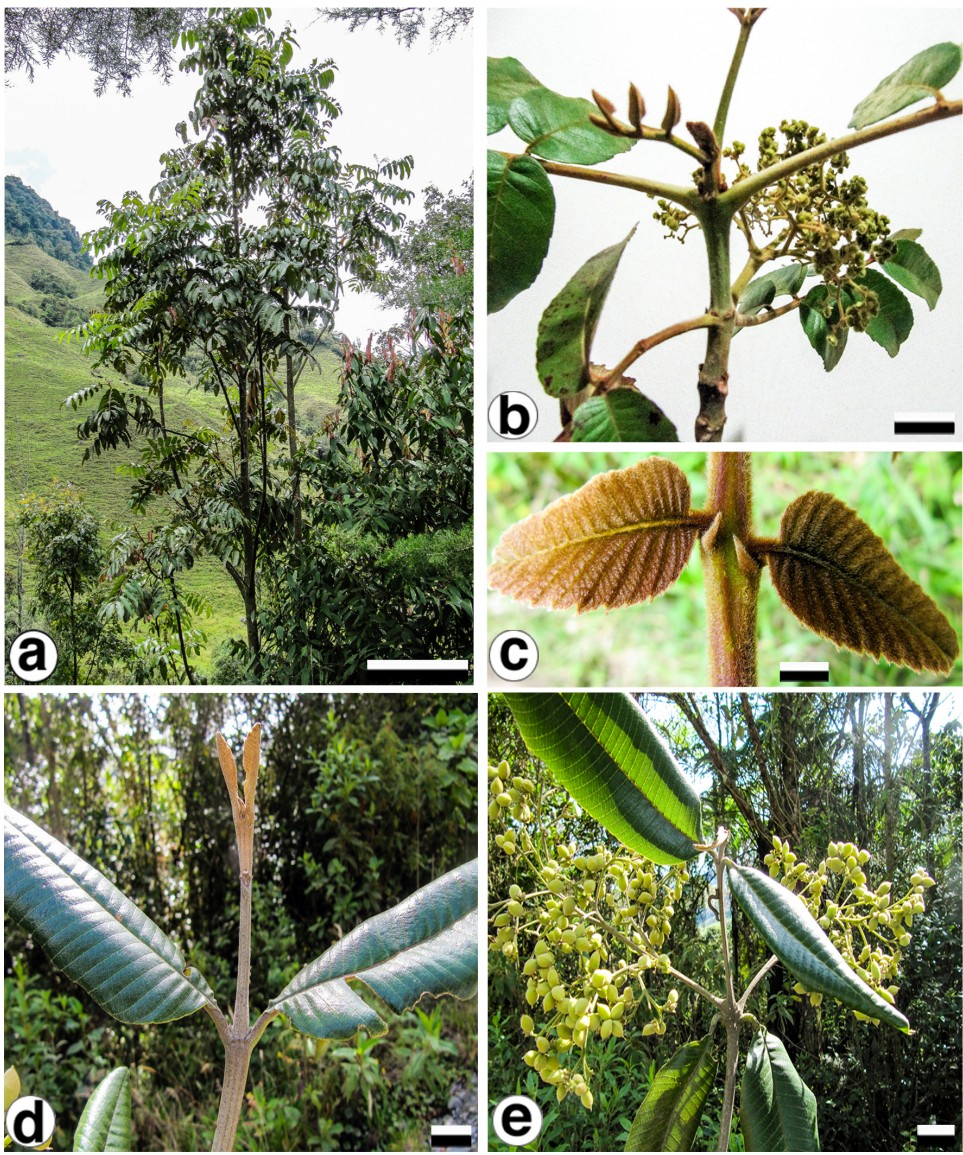

**Figure 1** *Brunellia antioquensis.* **(Cuatrec.) Cuatrec. and *B. trianae* Cuatrec.** *Brunellia antioquensis*: (A) habit, (B) twig with infructescences; (C) young leaflets and stipels. *Brunellia trianae*: (D) twig with adult leaves and vegetative buds; (E) twig with infructescences. (A)–(C) from *Orozco et al. 4021* (COL), (D)–(E) from *Orozco et al. 4019* (COL). Scale for (A), 2 m; (B), 2 cm; (C), 1 cm; (D)–(E), 2 cm. Photos by Clara I. Orozco.

were adjusted to 50 ng/µL to be sheared to fragments of approximately 500 bp. Library preparation, barcoding and sequencing on an Illumina HiSeqX were conducted at Rapid Genomics LLC (Gainesville, FL, USA).

## Plastid genome assembly and annotation

Sequencing process produced, on average, 7.5 Gb (s.d. = 0.2 Gb) reads per sample. Paired-end reads were trimmed (quality at 0.05 probability) and later assembled in Geneious 10.2.3

(Biomatters Ltd, Auckland, New Zealand). The assembly was performed by a combination of reference and *de novo* assemblies using as a reference the plastid genome of *Averrhoa carambola,* Oxalidaceae (GenBank accession KX364202). Annotations of gene-regions and rRNAs were transferred in Geneious from *Averrhoa* reference. The tRNAs were annotated using tRNAscan-SE v2.0 (*Lowe & Chan, 2016*) implemented in CHLOROBOX (https://chlorobox.mpimp-golm.mpg.de/geseq.html). All annotations were manually reviewed and, if necessary, edited. The circular map of the plastid genome was generated with OGDRAW (https://chlorobox.mpimp-golm.mpg.de/cite-OGDraw.html, *Lohse et al., 2013*) and modified manually.

## Comparison of plastid genomes of *Brunellia antioquensis* and *B. trianae*

Comparisons of the boundaries between the large single-copy (LSC) and the small single-copy (SSC) regions with the inverted repeats (IRA and IRB) were performed using Unipro UGENE v1.31.0 (*Okonechnikov, Golosova & Fursov, 2012*). Content percentages of A, T, C, G, A/T and G/C were estimated for LSC, SSC and IR regions, and for the rRNA, tRNA, protein coding regions, intergenic regions, and introns of both species with Bioedit v7.1.11 (*Hall, 1999*).

## Repeated sequences

Identification and location of complement, forward, palindromic, and reverse repeat sequences were conducted using REPuter program (*Kurtz et al., 2001*). Simple sequence repeats (SSRs) were identified using the MISA program with parameters set by default (*Thiel et al., 2003*).

## Comparative genome analysis

We compared *Brunellia* plastid genomes regarding nucleotide diversity ($\pi$), insertions/deletions (InDel) and base substitutions with the program DnaSP6 (*Rozas et al., 2017*). Transitions (Ts) and transversions (Tv) events were identified across all loci and for protein-coding regions, they were also classified based on the effect in the amino acid chain as synonymous (S) or non-synonymous (N) substitutions. Furthermore, the whole plastid genome sequences of *Brunellia* and eight other COM clade representatives were compared using Mauve (*Darling et al., 2004*) and mVISTA programs (*Frazer et al., 2004*). The eight sequences retrieved from GenBank are listed in Table S1. Plastid sequences of *Ceratopetalum apetalum, Sloanea australis* and *Tetraena mongolica* assembled by *Foster et al. (2016)* were annotated using the same procedure we implemented for the *Brunellia* samples.

## Phylogenomic analyses

We extracted and aligned by translation the protein-coding genes of the ten COM representatives in Geneious. We added 33 superrosid representatives (Table S1) from previously published matrices by *Ruhfel et al. (2014a)*, and pruned our data accordingly. The genes considered were *atpE-F, atpH-I, ccsA, cemA, clpP, infA, matK, ndhA-K, petA-B, petD, petG, petL, petN, psaA-C, psaI, psaJ, psbA-F, psbH-N, psbT, psbZ, rbcL, rpl2, rpl14,*

*rpl16, rpl20, rpl22-23, rpl32-33, rpl36, rpoA-B, rpoC1-C2, rps2-4, rps7-8, rps11-12, rps14-16, rps18-19* and *ycf2-4*. The *accD* gene alignment was removed from the final concatenated matrix due to its high variability that produced conflicts in the alignment. The *accD* has been reported as missing in species of Fabaceae (Fabids) and Geraniaceae (Malvids), and in some cases it has transferred to the nucleus (*Rousseau-Gueutin et al., 2013*; *Liu et al., 2016*). Phylogenetic analyses were performed using Maximum Likelihood using IQ-Tree web server (*Nguyen et al., 2014*; *Trifinopoulos et al., 2016*) with 1,000 ultrafast bootstrap replicates (*Hoang et al., 2017*) and the GTR+F+R3 model selected with ModelFinder under BIC (*Kalyaanamoorthy et al., 2017*).

### Information availability

The plastid genomes of *B. antioquensis* and *B. trianae* were prepared for submission for NCBI using GB2sequin (*Lehwark & Greiner, 2018*) and are available in the GenBank portal with the numbers MN585217 and MN615725. The nucleotide alignments are available in supplemental information.

## RESULTS

### Plastid genome features

A total of 73,003 for *B. antioquensis* and 97,054 for *B. trianae* paired-end reads were assembled to produce a mean coverage of 104.8× and 138.3×, respectively. The plastid genome size of *B. antioquensis* and *B. trianae* is 157,685 bp and 157,775 bp, respectively (Table 1) and consists of a single circular, double-stranded DNA sequence, showing the typical quadripartite structure (Fig. 2); both sequences include a pair of IR (26,376 bp in *B. antioquensis* and 26,389 bp in *B. trianae*), one SSC (17,538 bp in *B. antioquensis* and 17,542 bp in *B. trianae*), and one LSC (87,395 bp in *B. antioquensis* and 87,455 bp in *B. trianae*) (Table 1). The proportion of each region in the plastid genome for *B. antioquensis* is LSC 55.42%, SSC 11.12% and IRs 33.46%; for *B. trianae* these percentages are quite similar (Table 1). Content percentages of A, T, C, G, A/T and G/C are similar for both species (Table 1).

Both genomes consist of 52.13–52.14% protein-coding genes, 40.67–40.71% intergenic regions, 10.32–10.33% introns, 5.73% rRNAs, and 1.85% tRNAs (Table S2). Plastid genomes of *Brunellia* species have 134 coding regions, including 38 tRNAs, 8 rRNAs, and 88 protein-coding genes. Ninety-five genes are unique and 19 are duplicated in the IR regions (Table S3); of the 19 duplicate genes, seven are tRNAs, four are rRNAs, and seven are fully included and one interrupted protein-coding genes. Coding regions are distributed as follows: LSC included 22 tRNAs and 59 protein-coding regions, IRs have 4 rRNAs, 7 tRNAs, and 7 protein-coding regions, and SSC contains one tRNA and 11 protein-coding regions; additionally, *ycf1* is found between IRA and SSC border region, and *rps12* appears three times (twice in the IRs and once in the LSC region). Most of the protein-coding regions have one exon, 15 genes have one intron, whereas *clpP*, *rps12*, and *ycf3* present two introns (Table S4). Size of introns is between 544 to 2549 bp, being *trnK-UUU* the longer on the whole cpDNA. *Brunellia antioquensis* and *B. trianae* have similar IRA-SSC

**Table 1  Plastid genome characteristics of *Brunellia antioquensis* and *B. trianae*.**

| *B. antioquensis* | Size (bp) | Length (%) | A (%) | C (%) | G (%) | T (%) | C/G (%) | A/T (%) |
|---|---|---|---|---|---|---|---|---|
| LSC | 87,395 | 55.42 | 31.90 | 17.9 | 17 | 33.3 | 34.85 | 65.15 |
| IRB | 26,376 | 16.73 | 28.7 | 20.6 | 22.2 | 28.6 | 42.75 | 57.25 |
| SCC | 17,538 | 11.12 | 34.4 | 15.0 | 16.2 | 34.4 | 31.22 | 68.78 |
| IRA | 26,376 | 16.73 | 28.6 | 22.2 | 20.6 | 28.7 | 42.75 | 57.25 |
| Total | 157,685 | 100 | 31.08 | 18.73 | 18.36 | 31.83 | 37.09 | 62.91 |
| *B. trianae* | | | | | | | | |
| LSC | 87,455 | 55.43 | 31.91 | 17.86 | 16.98 | 33.25 | 34.84 | 65.16 |
| IRB | 26,389 | 16.73 | 28.7 | 20.6 | 22.2 | 28.6 | 42.75 | 57.25 |
| SCC | 17,542 | 11.11 | 34.4 | 15 | 16.2 | 34.4 | 31.23 | 68.77 |
| IRA | 26,389 | 16.73 | 28.6 | 22.2 | 20.64 | 28.7 | 42.75 | 57.25 |
| Total | 157,775 | 100 | 31.08 | 18.72 | 18.36 | 31.83 | 37.09 | 62.91 |

and SSC-IRB boundaries, but they are somewhat different by the number of base pairs at the LSC-IR borders (Fig. 3).

## Repeated sequences and SSRs

Twenty-six repeated sequences are found in the genome of *B. antioquensis*, whose lengths vary between 20 and 30 bp (Table S5). Twelve of those sequences are located in intergenic regions. Twelve repeats are palindromic, eight are forward repeats, three are complement repeats, and three are reverse repeats. Fourteen repeats are found in the LSC, six in the SSC, and four in the IR. Number, size and locations of repeated sequences in *B. trianae* are different (Table S6). This species has 18-repeated sequences with a length that varies between 21 and 88 bp; ten of these repeats are located in intergenic regions. Of the 18 repeats, six are palindromic, 11 are forward repeats, and one is a complement repeat; no reverse repeats are found in the cpDNA of this species. Thirteen repeats are found in the LSC, one in the SSC, and four in the IR regions.

The numbers of SSRs are quite similar between the two *Brunellia* species. *Brunellia antioquensis* has 223 SSRs, including 124 mononucleotides (66 poly T, 56 poly A, 1 poly C, 1 poly G), 19 dinucleotides, 69 trinucleotides, seven tetranucleotides, one pentanucleotide, one hexanucleotide, one 13-nucleotide, and one 21-nucleotide. *Brunellia trianae*, with 225 SSRs, differs from *B. antioquensis* by the number of dinucleotides (18) and the number of poly T (69). Dinucleotide SSRs are only represented by units containing TA/AT (82.4%) and TC/CT (17.6%) in both species.

## Analysis of polymorphisms among the plastid genome sequences of *Brunellia*

*Brunellia antioquensis* and *B. trianae* have very similar sequences. These species share 99.85% sequence similarity. In the analysis of the 157,844 bp aligned (gaps included), only 243 bp are different corresponding to 204 indels and 39 nucleotide substitutions (Table S7). The nucleotide variability expressed as the level of divergence among the sequences is $\pi = 0.00025$. An examination by sliding window of the alignment shows $\pi$ values range from 0 to 0.0067 for a 600 bp gradation (Fig. 4). Hotspots correspond

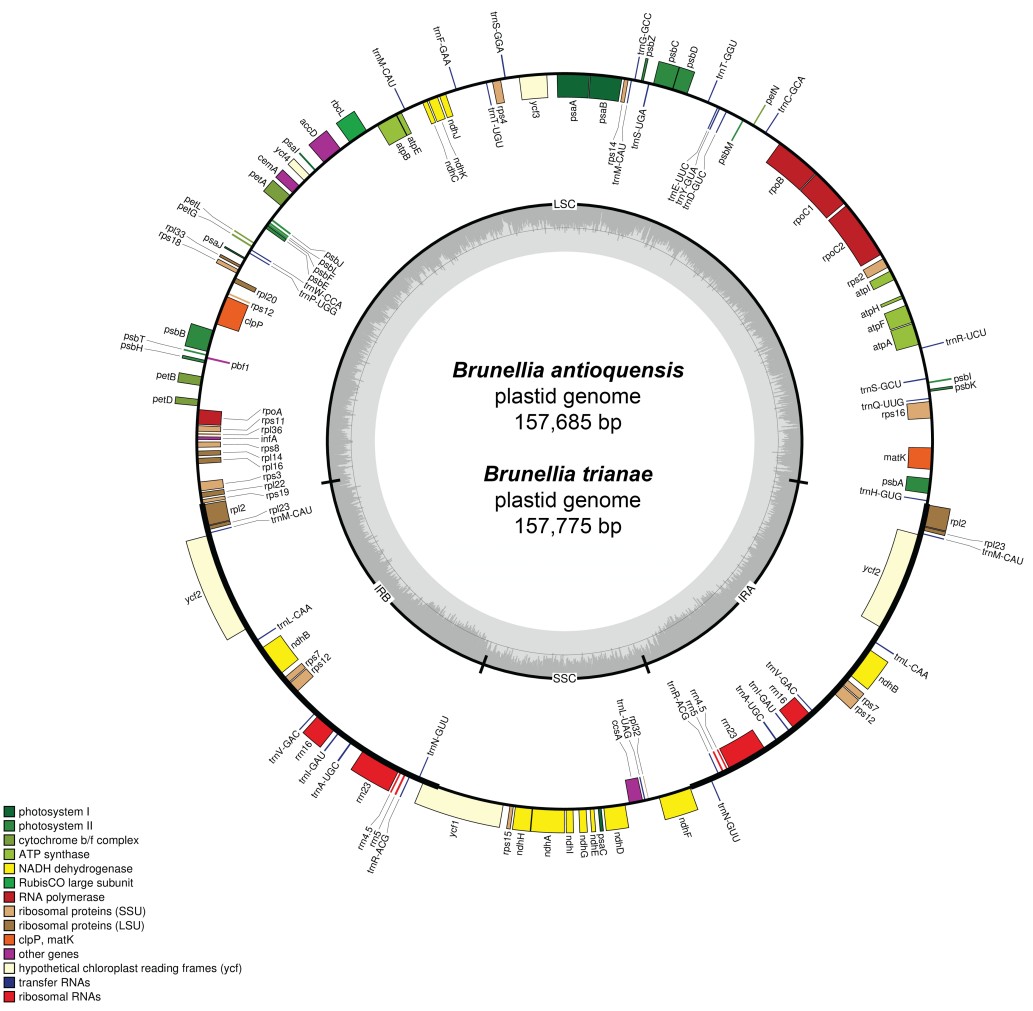

**Figure 2** **Gene map of the plastid genome of *Brunellia antioquensis* and *B. trianae*.** Genes placed inside of the outer circle are transcribed in counterclockwise direction whereas genes outside are transcribed in the clockwise direction. Colors refer to genes from different functional groups. The lighter grey area in the inner circle indicates AT content, while the darker grey area indicates GC content.

with the intergenic regions *rps2-rpoC2, petB-rpoA* and *ndhF-ccsA*. The SSC has the highest amount of substitutions per site ($\pi = 0.00068$), followed by the LSC ($\pi = 0.00026$); the IRs have the lowest value with only two substitutions in each ($\pi = 7.6 \times 10^{-5}$). Twenty-two transversions and 11 transitions evenly distributed among the regions of the genome are found, with a ratio of 2:1 of Tv to Ts. Fourteen substitutions occur in protein coding sequences of ten genes, including five transversions and nine transitions, giving a ratio of 1:1.8 of Tv to Ts. Only five substitutions are synonymous, four of them due to the replacement of thymine (T) by cytosine (C).

Among the sequences, two thirds of the indel events (20) involve the gain or loss of single nucleotides of adenine (A) or thymine (T), and they are associated with regions where they appear repeated more than six times. Twenty-four indels are located in intergenic regions,

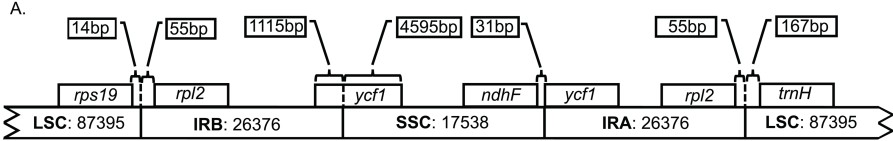

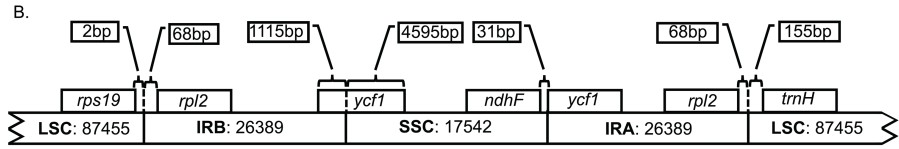

**Figure 3** **Boundaries among different plastid regions.** (A) *Brunellia antioquensis.* (B) *B. trianae.* Apparent size of each region, gene, and distance in bp at ends (not to scale).

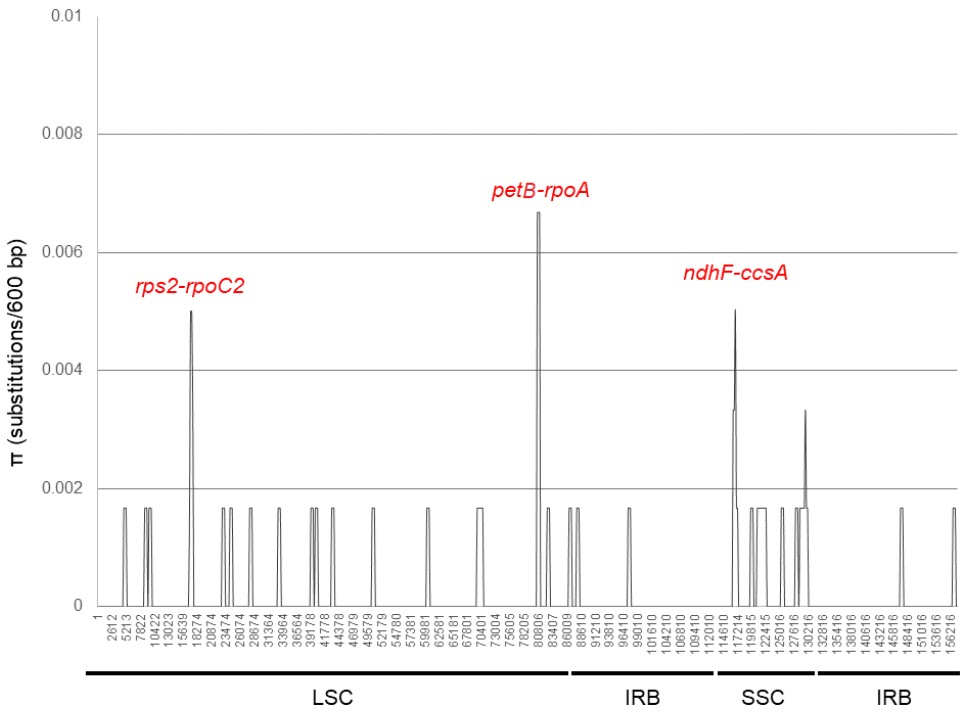

**Figure 4** **DNA polymorphism analysis of the plastid genome sequences of *Brunellia antioquensis* and *B. trianae.*** Nucleotide diversity ($\pi$) evaluated with DnaSP6 (settings: window length: 600 bp, step size: 200 bp).

six indels in introns and one in the *rpoC2* exon. This last indel creates a change in nine nucleotide sites and causes three amino acid changes. The longest indel was 83 bp long and is located in the LSC region between *petN* and *psbM*.

## Comparison of the plastid genomes of COM clade representatives

Three Locally Collinear Blocks are identified as the LSC, IR, and SSC (Fig. S1). The possible lack of information of the *Sloanea australis* plastid genome in the regions adjacent to the

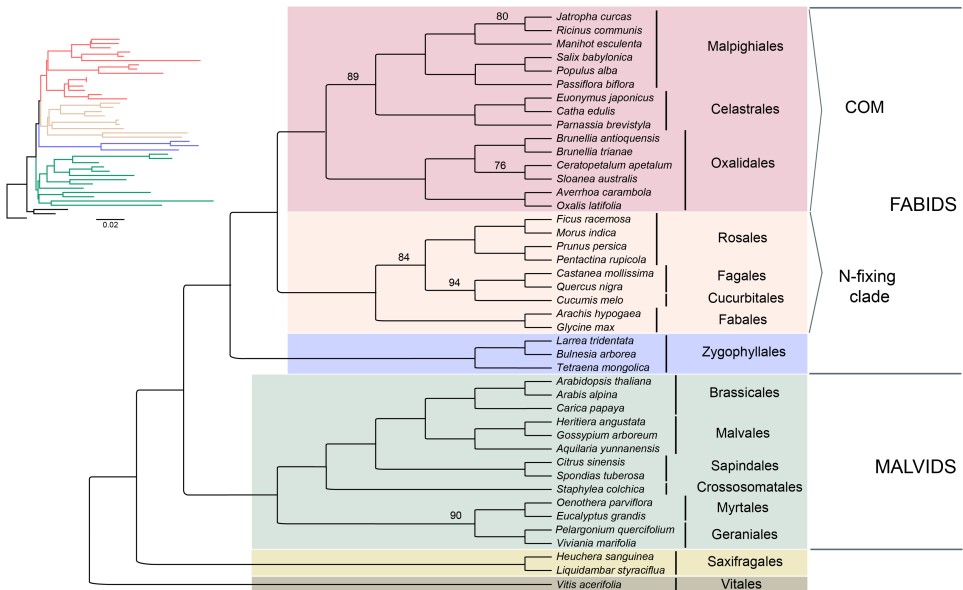

**Figure 5  Phylogenetic relationships of 43 superrosid representatives.** Relationships inferred from the 75-plastid protein-coding regions using maximum likelihood (ML) analysis. Bootstrap values are 100% except where indicated.

LSC-IR and IR-SSC junctions may be the reason why there is not one longer collinear block. This is also detectable in the Mauve output as two inversions in the LSC and SSC. Those apparent inversions denote a different starting point for the reported *Sloanea* sequence, which might be caused by a possible problem with the assembly. In the mVISTA analysis (Fig. S2), using *B. antioquensis* as a reference, there is high similarity among *Brunellia* and other Oxalidales plastid genomes.

## Phylogenomic analyses

Seventy-five protein-coding regions were aligned in a matrix of 54,367 bp length. ML analysis places Brunelliaceae in a clade with Cunoniaceae and Elaeocarpaceae, where Oxalidaceae is sister to these 3 families (Fig. 5). The monophyly of the COM clade, as well as Fabids and Malvids, is recovered with 100% bootstrap support (BS). Oxalidales appear as sister group of the Celastrales-Malpighiales clade with 89% BS. The sister group of the COM clade is the N-fixing clade, and Zygophyllales is sister to them with 100% BS in both cases. The Malvid clade is sister to Fabids and is supported by 100% BS.

## DISCUSSION

We report the first two plastid genomes for Brunelliaceae that are added to the previously published data for Oxalidales: the fully annotated plastid genome of *Averrhoa carambola* (*Jo et al., 2016*), the preliminary plastid sequences of *Ceratopetalum apetalum* and *Sloanea australis* (*Foster et al., 2016*) and the plastid regions of *Oxalis latifolia* (*Moore et al., 2010*). Structure, organization, and AT/GC content are similar to the *A. carambola*, COM group, and other Eudicot plastid genomes (e.g., *Ravi et al., 2008*).

Although *B. antioquiensis* and *B. trianae* have different morphological characters in their leaves, indument and fruits, the plastid sequence of them is surprisingly similar. *Brunellia trianae* has a plastid genome that is only 90 bp longer than *B. antioquensis* (Table 1). Differences among the size of the sequences have been related to the expansion and contraction of the IR regions (*Downie & Jansen, 2015*), but in *Brunellia* it does not seem to be the case; variations of the boundaries occur only between the LSC region and the IRs (Fig. 3), and at these boundaries the variations are small (12 to 13 nucleotides). Therefore, IR contractions or expansions do not affect the size of the genome in the two species studied. Plastid genome variation size between species is mostly explained by an indel of 88 bp located in the LSC region (between *psbM* and *petN*) in *B. antioquensis*. Similar results were found in *Tilia*, another rosid with low variability among congeneric genomes (*Cai et al., 2015*).

Variations between *B. antioquensis* and *B. trianae* comprise a difference of 39 nucleotides substitutions and 204 indels. The sequences show relatively low nucleotide variability ($\pi = 0.00025$), with IRs less variable ($\pi = 7.6 \times 10^{-5}$) as expected due to their effect in keeping the structure and stabilization of the chromosome (e.g., *Sun et al., 2016a*; *Sun et al., 2016b*; *Wicke et al., 2011*). The nucleotide variability found among *Brunellia* species is also one of the lowest reported. In general, the $\pi$ values in trees tend to be lower than in herbs; for example, in trees: *Machilus* ($\pi = 0.00154$, *Song et al., 2015*) and *Juglans* ($\pi = 0.00219$, *Hu, Woeste & Zhao, 2017*); in herbs: *Nicotiana* ($\pi = 0.00321$, *Asaf et al., 2016*), *Aconitum* ($\pi = 0.00549$, *Kong et al., 2017*) and *Papaver* ($\pi = 0.00895$, *Zhou et al., 2018*). Discrepancies in the rate of molecular substitutions can be explained by variations of generation time, size and habit, the age of the clade, and molecular mechanisms of DNA repair, among others (e.g., *Lanfear et al., 2013*; *Smith & Donoghue, 2008*). In the case of *Brunellia* trees, likely rapid and recent diversification in the last 5 mya (José Murillo-A, Clara Inés Orozco, Carlos Parra-O, Alvaro J. Perez and Katya Romoleroux, 2019, unpublished data) might explain a $\pi$ value ten times smaller than the one recorded for other trees. High morphological diversity and low molecular divergence has also been found in other Andean clades that speciated during the Plio-Pleistocene (e.g., *Hughes & Eastwood, 2006*; *Nürk, Scheriau & Madriñán, 2013*).

Previously, we analyzed eight plastid regions and found low variability and no phylogenetic signal (José Murillo-A, Clara Inés Orozco, Carlos Parra-O, Alvaro J. Perez and Katya Romoleroux, 2019, unpublished data). This fact is not surprising, given the overall low variability in the complete plastid genome. In this study we found three variable intergenic regions (*rps2-rpoC2*, *petB-rpoA* and *ndhF-ccsA*, $\pi = 0.005 - 0.0067$); although with low sequence divergence, they could represent the most variable plastid regions useful in phylogenetic analysis of *Brunellia*.

The use of either complete plastid genomes or many coding regions have proved to increase the resolution and support of the relationships in phylogenetic analyses (e.g., *Jansen et al., 2007*; *Moore et al., 2007*). In our analyses, the inclusion of *Brunellia* sequences, the data produced by *Foster et al.* (*2016*; *Sloanea australis* and *Ceratophyllum apetalum*), and the plastid genomes published by *Jo et al.* (*2016*; *Averrhoa carambola*), *Gu et al.* (*2018*; *Catha edulis*) and *Xia et al.* (*2018*; *Parnassia brevistyla*) gave us the opportunity to improve

the dataset for Oxalidales and Celastrales in terms of amount of characters and taxon sampling. Despite the possible assembly issues in the *S. australis* plastid genome, the coding regions placed it in the expected phylogenetic position, so we decided to include it in the complete analysis. To date, plastid sequences of *Averrhoa* and *Oxalis* were the only ones used to represent the whole order (*Sun et al., 2015*; *Jo et al., 2016*).

The phylogenetic reconstruction placed Brunelliaceae in the clade Oxalidales and sister to the clade Cunoniaceae-Elaeocarpaceae with 76% BS, which is congruent with *Soltis et al. (2011)*. These relationships disagree with *Zhang & Simmons (2006)* in which Brunelliaceae is sister to Cunoniaceae. *Sun et al. (2016a)*, *Sun et al. (2016b)* and *Heibl & Renner (2012)* include in their analyses Cephalotaceae (not present in our sampling) and do not recover the clade Cunoniaceae-Elaeocarpaceae. In these, Brunelliaceae is sister to Cephalotaceae, this clade sister to Elaeocarpaceae, and Cunoniaceae sister to all three. Further work is needed with a wider sampling to disentangle the phylogenetic relationships among these four families.

Our results depict that the COM clade is monophyletic with 100% BS, which is consistent with many phylogenetic analyses from plastid data (see Table S8). However, nuclear data sets have not shown this monophyly, and support the relationships (C,M)(O,Malvids p.p.) (*Zhao et al., 2016*; *Zeng et al., 2017*; *Leebens-Mack et al., 2019*). Inside the COM clade, the relationships among the three orders have been conflicting. Phylogenetic analyses that support each one of the three possible topologies among the orders are compiled in Table S8. We found that Celastrales and Malpighiales form a clade with 89% BS making the ((C,M)O) topology the most likely according to our data. These relationships had been found by *Li et al. (2019)* using 80 plastid genes, by *Cauz-Santos et al. (2017)* using 43 plastid coding regions, by *Moore et al. (2011)* using all IR data, and by *Zhang & Simmons (2006)*, *Soltis, Gitzendanner & Soltis (2007)* and *Burleigh, Hilu & Soltis (2009)* using few plastid and nuclear genes. The (C(O,M)) topology was found by *Ruhfel et al. (2014b)* using 1,2 codon positions and *Wu et al. (2014)* using all three, however both show low bootstrap support. The third possible topology ((C,O)M) has been also recovered using plastid coding regions (*Ruhfel et al., 2014b*; *Sun et al., 2015*), but the change in the taxon sampling leads to the ((C,M)O) topology (Fig. 5). Therefore, conflicting results with other plastid-based phylogenies may be caused by the small sampling of Celastrales (i.e., *Euonymus americanus*) and Oxalidales (i.e., *Oxalis latifolia* or *Averrhoa carambola*) in the analyses (*Ruhfel et al., 2014b*; *Wu et al., 2014*; *Sun et al., 2015*; *Jo et al., 2016*) or by the use of an amino acid dataset (*Ruhfel et al., 2014b*; *Gitzendanner et al., 2018*).

The phylogenetic analysis shows that the N-fixing clade is sister to the COM clade with 100% BS. This relationship has been found in other studies based on plastid data (*Sun et al., 2015*, Table 1). However, when nuclear and mitochondrial data are analyzed the COM clade is sister to Malvids or even not monophyletic. *Sun et al. (2015)* developed a hypothesis of the possible origin of the COM clade genome by the action of introgression and hybridization processes from both malvid and fabid ancestors. This scenario, in which the mitochondrial and most of the nuclear genomes would have their origin from a Malvid ancestor, and the plastid genome from a Fabid ancestor, would explain the observed discrepancies in the phylogenies (*Sun et al., 2015*). Therefore, deep analyses based on molecular data from all

three genomes will give us a better understanding of the evolutionary processes within the diversity of rosids.

Based on our analysis of plastid data, the Fabid clade appears monophyletic (including Zygophyllales) with 100% BS. Zygophyllales is sister to the rest of Fabids, as has been previously found with large datasets (e.g., *Soltis et al., 2011*; *Sun et al., 2015*), but further analysis including more representatives of Zygophyllales is necessary because other studies have found the order sister to Malvids (*Zhao et al., 2016*) or in an uncertain position (*Ruhfel et al., 2014b*).

## CONCLUSIONS

Assemblies of complete plastid genomes have proved to be useful for the identification of variable sites that can work as barcodes for specific groups; however, *Brunellia* sequences do not show high levels of variation and are not particularly useful for this purpose. Our results agree with previous analyses in which we tested eight plastid regions for phylogenetic purposes. Here, we present for the first time, plastid genomes for Brunelliaceae. We analyzed the genomes of two species to look for regions with high variability, but we found a very low level of nucleotide diversity. The three main hotspots identified among them were rich in indel regions, with some potential in further systematic analyses. Phylogenetic analyses of *Brunellia* plastid regions plus another 41 superrosid representatives corroborate the placement of the family in Oxalidales, and as sister group of a clade formed by Cunoniaceae and Elaeocarpaceae. Our data also support Oxalidales as sister to the Celastrales-Malpighiales clade. This topology disagrees with the most common scenario of Celastrales and Oxalidales as sister groups; however, all three possible topologies have been found with both small and large datasets. Possible causes of incongruence may be related to taxon sampling and the lack of information for many species in the less diverse orders, as Celastrales and Oxalidales, which tend to be poorly represented in the phylogenies. Future studies focused on the COM group should include a better representation of families and genera in order to understand other evolutionary trends in this relatively recently recognized clade.

## ACKNOWLEDGEMENTS

We would like to thank Jorge Vélez and Juan Pablo Tobón at the Universidad Nacional de Colombia Sede Medellín, for assisting with fieldwork.

### Funding

This work was supported by the "Departamento administrativo de ciencia, tecnología e innovación" (Colciencias-1101658) grant to Clara I. Orozco, Jose Murillo-A, and Carlos Parra-O; the "División de investigación y extensión" of Universidad Nacional de Colombia (DIEB-15858) grant to Clara I. Orozco and Jose Murillo-A; the Southern Illinois University (SIU) for start up funds and support from NSF DUE-1564969 to Kurt M. Neubig; the

Fulbright-Colciencias scholarship and the SIU Cope Fund grant to Janice Valencia-D. The funders had no role in study design, data collection and analysis, decision to publish, or preparation of the manuscript.

### Grant Disclosures

The following grant information was disclosed by the authors:
Departamento administrativo de ciencia, tecnología e innovación: 1101658.
División de investigación y extensión: DIEB-15858.
Southern Illinois University.
NSF: DUE-1564969.
Fulbright-Colciencias scholarship and the SIU Cope Fund.

### Competing Interests

The authors declare there are no competing interests.

### Author Contributions

- Janice Valencia-D conceived and designed the experiments, performed the experiments, analyzed the data, prepared figures and/or tables, authored or reviewed drafts of the paper, made the bioinformatic assembly of the data, and approved the final draft.
- José Murillo-A conceived and designed the experiments, analyzed the data, prepared figures and/or tables, authored or reviewed drafts of the paper, and approved the final draft.
- Clara Inés Orozco and Carlos Parra-O conceived and designed the experiments, authored or reviewed drafts of the paper, and approved the final draft.
- Kurt M. Neubig conceived and designed the experiments, performed the experiments, authored or reviewed drafts of the paper, and approved the final draft.

### Field Study Permissions

The following information was supplied relating to field study approvals (i.e., approving body and any reference numbers):

The samples used in this study were collected under the institutional Universidad Nacional de Colombia collection permit. (Permit Number: 0255).

### DNA Deposition

The following information was supplied regarding the deposition of DNA sequences:
Complete plastid sequences are available at Genbank: MN585217 and MN615725.

### Data Availability

The plastid genomes are available in the Supplemental Files.

### Supplemental Information

Supplemental information for this article can be found online at http://dx.doi.org/10.7717/peerj.8392#supplemental-information.

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
