# Peer review of "­Complete plastid genome sequences of two species of the Neotropical genus Brunellia (Brunelliaceae)"

_PeerJ, doi:10.7717/peerj.8392_

## Round 0.1 · original submission · Minor Revisions

Your manuscript received "good grades" from the 3 reviewers. There are several minor issues to address. Your abstract needs to contain motivation and the research goal. Make sure to cite all major paper in this area (some are suggested by reviewers). One important aspect to address is a possible bias of the sample. Another is not to overstate the importance of plastid genomes and explain the limitations of their usability for phylogeny.

Reviewer 1 ·

Basic reporting

No comment.

Experimental design

No comment.

Validity of the findings

No comments.

Additional comments

It is normal paper to present the plastomes. It is first paper that present plastome of Brunelliaceae. The features of plastome of Brunelliaceae were also analyses. The phylogenetic position of Brunelliaceae were also inferred.
It would be better to inlude some photos of two species in the paper to illustrate the morphological difference to these two species.

Reviewer 2 ·

Basic reporting

A:
Page(P) Line(L) 111, two species names are not italicized

P10L210 genus name is not italicized

B: these two papers should be cited for ((C,M)O) topology
Soltis et al. 2007 A 567-taxon data set for angiosperms The challenges posed by Bayesian analyses of large data sets

Exploring the phylogeny of rosids with a five-locus supermatrix from GenBank
Miao Sun, Ryan A. Folk, Matthew A. Gitzendanner, Stephen A. Smith, Charlotte Germain-Aubrey, Robert P. Guralnick, Pamela S. Soltis, Douglas E. Soltis, Zhiduan Chen
bioRxiv 694950; doi: https://doi.org/10.1101/694950

Experimental design

A:
Based on the abstract, the motivation and research goal is not stated, even though I read the answers in the introduction part.

B:
Since the phylogenetic framework of the flowering plants has been established by plastid genome data, I and maybe some readers are really curious why authors put so much efforts in the plastid genome, and just for two species?

C:
There are only 11 sequences in INFA.fasta file, far below the average; Including this alignment in the supermatrix would introduce extra gaps, could bias the model estimation and phylogenetic tree inference.

Validity of the findings

A:
I feel like this work is solid but lacking of novelty and similar phylogeny from plastid/cp gene/genomes have been widely reported.


B:
The authors reconstructed phylogeny using 75 plastid protein-coding regions for 41 selected superrosid species. How this gene and taxon sampling schema is better than studies from (Ruhfel et al., 2014; 2014; Sun et al., 2015; etc). If the COM clade is heavily sampled, for 41 species, that means the rest clade of superrosid are sparse, which also means biased sampling, isn't it? Additionally, for three genomics in plant (mitochondrial, plastid, and nuclear), only sampled plastid is also not sufficient to resolve the inter relationship of OMC clade.

Additional comments

This manuscript is general well written, data is solid and analyses are statistically sound; all done in a start genomics and phylogenomic manner, however, I could not find the whole research motivation and results are exciting and appealing. Moreover, I think the authors should also justify their gene and taxa sampling is not biased. The era of application of plastid genome has passed, the whole tree of life in plant has been well established plastid genome, now the community should move on march to the biparental, more informative nuclear genes. Eventually plastid genome only tells one side of evolutionary story.

·

Basic reporting

I almost never review a manuscript so ready for publication. It is is clearly written and well explained.

Experimental design

This work has been done in accordance with the highest standards and best practices.

Validity of the findings

I have no doubts about the validity of these findings.

Additional comments

I have only these minor suggestions:

(1) Throughout, these authors use the past tense to describe observations that are currently true (although discovered in the past during the study), e.g., “Plastid genome size of B. antioquensis and B. trianae was 157,685 bp and 157,775 bp, respectively.” Although this is not uncommon in scientific writing, I personally judge that the present tense is more appropriate (while retaining past tense where appropriate, as when referring to something being done for the study, of course). This is just my personal preference and the authors are free to decide whether that would improve what is generally excellent writing.

(2) Line 201: Is the word “Brunellia” extraneous here?

(3) Lines 224 and 225: The text reads, “All COM sequences analyzed had the same gene order and three Locally Collinear Blocks were 225 identified in the LSC, IR, and SSC (Fig. S1).” It is not clear at this point what taxa share this “Locally Collinear Blocks,” so perhaps this should be preceded by the listing of the other COM species from the literature are included for this statement.

(4) Table 1: I suggest replacing “C/G” with “C+G” and “A/T” with “A+T.”

---

## Round 0.2 · Minor Revisions

You manuscript has been greatly improved. There is, however, a list of minor suggestions that I encourage you to consider. The second reviewer has also provided the annotated manuscript with comments. Please take it into account when working on your revision.

Reviewer 1 ·

Basic reporting

This revision is in good condition. Most of problem has been resolved. The language is professional.

Experimental design

Good. It falls within the PeerJ.

Validity of the findings

Good.

Additional comments

The manuscript is ready for acception.

Reviewer 2 ·

Basic reporting

As stated in my first review

Experimental design

As stated in my first review

Validity of the findings

As stated in my first review

Additional comments

Update review comments based on authors' revision:

The authors did a great job in their revision. I'm also glad to see the authors conducted additional analyses by excluding INF A region, and compared with their original results, and confirmed the results are not impacted.

I'm basically fine with this ms as it is, except one case as detailed below:

P10, L335-342: The authors stated that "Thus, the position of the COM clade remains uncertain (APG 2016).

1. I thought the phylogentic placement of COM clade has been nailed: COM clade is embedded in malvids (maybe not monophyletic). Please check out recent 1kp paper (One Thousand Plant Transcriptomes Initiative, 2019): COM clade is nest in malvids in their ASTRL species tree analysis (Please see summarized rosid tree in the attached pdf).


2. Therefore I feel that this conclusion is out of date, and after APG IV (2016), there are so many studies pop out using nuclear data (e.g., Zeng et al., 2017; Leebens-Mack et al., 2019; Smith et al., 2019).

3. The authors should be clear that the Fabid clade is monophyletic only based on plastid data. Because it's not monophyletic, as nuclear data suggests COM clade and Zygophyllales are nested in malvids, respectively (Please see summarized rosid tree in the attached pdf). Please see Leebens-Mack et al. (2019) and Smith et al. (2019)

4. Thus I suggest authors should clearly declare in their results, discussion and conlusion sections that statements in present study only represent plastid data only (one side story from plant plastid genome). I know this study is only based on plastid data discussing phylogenetic relationships among taxa in rosids (as the authors fairly explained in the rebuttal letter ). However, the authors should acknowledge the new findings from nuclear datasets regarding to their focal taxa. The phylogenetic relationships are hard to be resolved if we only relied on data/information from only one part of three plants genomes.

FYI:
One Thousand Plant Transcriptomes Initiative, 2019:
Leebens-Mack, J.H., Barker, M.S., Carpenter, E.J. et al. One thousand plant transcriptomes and the phylogenomics of green plants. Nature 574, 679–685 (2019) doi:10.1038/s41586-019-1693-2

Smith et al., 2019:
Stephen A Smith, Nathanael Walker-Hale, Joseph F Walker, Joseph W Brown, PHYLOGENETIC CONFLICTS, COMBINABILITY, AND DEEP PHYLOGENOMICS IN PLANTS, Systematic Biology, , syz078, https://doi.org/10.1093/sysbio/syz078

Zeng L, Zhang N, Zhang Q, Endress PK, Huang J, Ma H. 2017. Resolution of deep eudicot phylogeny and their temporal diversification using nuclear genes from transcriptomic and genomic datasets. New Phytologist 214: 1338–1354.


One small thing:

In "SUPLEMENTARY MATERIAL" file, the genus name "Brunellia" should be italic in the first title

Annotated reviews are not available for download in order to protect the identity of reviewers who chose to remain anonymous.

---

## Round 0.3 · accepted · Accept

I feel that the authors have addressed the concerns of the reviewers and the manuscript is in good shape for the publication.